# The Links between Microbiome and Uremic Toxins in Acute Kidney Injury: Beyond Gut Feeling—A Systematic Review

**DOI:** 10.3390/toxins12120788

**Published:** 2020-12-11

**Authors:** Alicja Rydzewska-Rosołowska, Natalia Sroka, Katarzyna Kakareko, Mariusz Rosołowski, Edyta Zbroch, Tomasz Hryszko

**Affiliations:** 12nd Department of Nephrology and Hypertension with Dialysis Unit, Medical University of Białystok, 15-276 Białystok, Poland; nmasalska1@wp.pl (N.S.); katarzyna.kakareko@umb.edu.pl (K.K.); edyta.zbroch@umb.edu.pl (E.Z.); tomasz.hryszko@umb.edu.pl (T.H.); 2Department of Gastroenterology and Internal Medicine, Medical University of Białystok, 15-276 Białystok, Poland; mariusz.rosolowski@umb.edu.pl

**Keywords:** acute kidney injury, microbiota, uremia middle molecule toxins

## Abstract

The last years have brought an abundance of data on the existence of a gut-kidney axis and the importance of microbiome in kidney injury. Data on kidney-gut crosstalk suggest the possibility that microbiota alter renal inflammation; we therefore aimed to answer questions about the role of microbiome and gut-derived toxins in acute kidney injury. PubMed and Cochrane Library were searched from inception to October 10, 2020 for relevant studies with an additional search performed on ClinicalTrials.gov. We identified 33 eligible articles and one ongoing trial (21 original studies and 12 reviews/commentaries), which were included in this systematic review. Experimental studies prove the existence of a kidney-gut axis, focusing on the role of gut-derived uremic toxins and providing concepts that modification of the microbiota composition may result in better AKI outcomes. Small interventional studies in animal models and in humans show promising results, therefore, microbiome-targeted therapy for AKI treatment might be a promising possibility.

## 1. Introduction

The term microbiota is used to describe a community of 100 trillion microorganisms (more than 1000 species, of mostly bacteria, but also viruses, fungi, and protozoa) that are present in the gastrointestinal tract. The term microbiome on the other hand, refers to the collective genomes of the above-mentioned organisms. The microbiome encodes over 3 million genes (as compared to 30,000 genes present in the human genome) characterized now by the Human Microbiome Project [1], which produce thousands of metabolites. Unsurprisingly, those metabolites play an important role both in human biology and disease development [2].

In healthy subjects, the phyla *Bacteroidetes* and *Firmicutes* dominate, composing more than 90% of all species, although density and species composition varies alongside the digestive tract [3]. The functional diversity of microbiota is key to normal biology.

Kidney failure directly or indirectly modifies microbial composition in the digestive tract. Edema and ischemia of the intestinal wall complicating kidney diseases lead to increased intestinal permeability, or so-called “leaky gut”, which further activates the immune system and results in systemic inflammation.

The relationship is bidirectional, as dysbiosis (changes in structure and composition of the microbiome) is implicated in many kidney diseases. The existence of a gut-kidney axis has been extensively proved. Communication between those two organs occurs not only directly, but also through the aforementioned metabolites. We have long known that many uremic toxins, such as ammonia, indole sulfate, p-cresyl sulfate, indole-3 acetic acid, trimethylamine N-oxide (TMAO) are gut-derived. Short chain fatty acids (SCFAs)–anaerobic fermentation end products of complex carbohydrates, on the other hand, are insufficiently generated due to dysbiosis. Most research focuses on chronic kidney disease (CKD) and the unmistakable role these toxins play in cardiovascular complications, mortality, or disease progression [4]. Less is known about the gut-kidney crosstalk present in acute kidney injury (AKI).

## 2. Objective

This systematic review aims to answer questions about the role of microbiome and gut-derived toxins in acute kidney injury. We analyzed both experimental and interventional data, animal and human studies, original articles and reviews/opinions aiming to gather full knowledge on the subject in question.

## 3. Results

Out of 1014 identified records, 638 were screened, 36 retrieved and assessed for eligibility and 33 finally included in the review (21 original studies and 12 reviews/commentaries). The entire selection process is illustrated in Figure 1. The basic characteristics of the original articles on the researched subject are summarized in Table 1 and Table 2. We have identified 16 experimental studies and five human studies (three prospective observational and two retrospective). We could therefore not assess the risk of bias and applicability concerns. ClinicalTrials.gov identified 226 studies with 36 duplicates, and after screening, one study, pertaining to the subject of the review remained.

## 4. Discussion

Acute kidney injury is usually multifactorial and involves hemodynamic, biochemical, inflammatory, and immunological mechanisms. The first studies on the influence of microbiome on AKI stemmed from the “hygiene hypothesis”, which states that the lack of exposure to normal microbiota induces less immune deviation and reduces immune regulation [26]. Jang and coworkers studied the effects of ischemia-reperfusion injury (IRI) on germ-free mice and found that compared to controls, the severity of injury (measured with creatinine and histologically determining the extent of tubular injury) was higher. After addition of bacteria to the diet, renal injury was equivalent to that of control mice [5].

### 4.1. The Gut-Kidney-Axis

There is a multitude of evidence that CKD alters microbiome composition [27]. Experimental animal evidence suggests that such is also the case with acute kidney injury. The relationship is bidirectional—AKI not only causes dysbiosis, microbiota composition also determines the severity of kidney injury [6,9,10]. In a murine model, hypoxia altered the ratio of aerobic/anaerobic population, increased the quantity of strict anaerobes and total lactic acid bacteria [6]. In another experimental setting of ischemia reperfusion injury, the balance between *Rothia* and *Streptococcus* genera was associated with creatinine levels [10]. 

Gut microbiota affect not only the levels of certain uremic toxins but both adaptive and innate immune responses. The numbers of natural killer T cells were higher in germ-free mice, AKI induction resulted in a greater number of CD8 cells [5]. Th17 cells are also crucial in kidney injury, as they produce Il-17 and directly induce renal inflammation by neutrophil activation and macrophage-mediated injury. Emerging evidence shows that the immunomodulatory effects of the intestinal microbiota include Th17 induction [28].

Another key link between the gut and the kidney is immunoglobulin A (IgA). It limits bacterial association with the epithelium [29] and prevents bacterial penetration of host tissue [30]. The exact mechanisms by which it works still remain unclear. Clues from research on IgA nephropathy also point to existence of an IgA axis between the mucosa, the bone marrow, and the kidney. In that specific disease, genome-wide association studies have identified risk loci in genes involved in the maintenance of the intestinal epithelial barrier and response to mucosal pathogens. The genetic risk also strongly correlates with microbiota variation, particularly helminth diversity [31]. The role of IgA in acute kidney injury is still unknown.

Drugs used for kidney disease treatment also affect the gut-kidney axis. One of the examples is oral iron, which increases gut microbial protein fermentation and in turn causes an increase in fecal and plasma uremic toxin levels. Increase in gut transit time may further increase the levels of many uremic toxins (such as phenols and indoles) [32,33]. The effect of many drugs used for AKI treatment on microbiome remains largely unknown.

As all intestinal barriers, biological, physical, and immune barriers might become damaged during kidney injury (the aforementioned “leaky gut hypothesis”), as levels of other toxins, mainly inflammatory, rise. Circulating lipopolysaccharide and endotoxin levels are higher during AKI and stimulate intrarenal inflammatory response [8].

A schematic review of the gut-kidney axis in AKI is shown in Figure 2.

### 4.2. Uremic Toxins

Healthy kidneys excrete a multitude of substances. In patients with CKD retention, these substances occur. These substances, when they contribute to the development of uremia, are called uremic toxins. They can be divided into three categories: small water-soluble, non-protein-bound molecules (such as urea, guanidines, oxalate, TMAO, phosphorus), small, lipid-soluble or protein-bound molecules (such as p-cresol sulfate, indoles, homocysteine), and the so-called middle molecules such as beta-2 microglobulin. It is well-known that some of these toxins are produced by the microbiota. Dysbiosis leads to excessive secretion of gut-derived uremic toxins, which in turn may further damage renal tubular cells [34]. 

Experimental evidence shows that selective modification of the microbiome might change the uremic toxin profile, e.g., in mice, deleting the tryptophanase gene from *Bacteroidetes* present in the digestive tract resulted in nearly undetectable indole sulfate levels [35]. Both indole sulfate and p-cresyl sulfate are well-researched toxins and their levels correlate with cardiovascular events and mortality in dialysis patients [36,37]. The levels of those toxins rise in acute kidney injury and correlate with RIFLE classification of AKI severity [15]. In a prospective cohort study, serum indoxyl sulfate levels were also associated with mortality in hospital-acquired AKI [14]. In a murine model, suppression of indole and p-cresyl production by *Lactobacillus salivarius* protected against cisplatin-induced kidney injury [23].

Trimethylamine-N-oxide is a gut-derived metabolite of phosphatidylcholine and a known uremic toxin. It directly enhances atherogenesis. There is a proven correlation between elevated TMAO levels and major adverse cardiovascular events (death, myocardial infarction, or stroke) [38]. CKD patients (including hemodialysis) with elevated TMAO levels have lower long-term survival [39,40]. Studies in AKI on TMAO are still pending.

Homocysteine (Hcy), a transmethylation product of metabolic conversion of methionine to cysteine, is an extensively studied gut-derived uremic toxin. Hyperhomocysteinemia is implicated in the progression of chronic kidney disease, and many cardiovascular complications [41,42]. In an experimental murine model, it induced more severe cisplatin induced-AKI than in mice with normal Hcy levels [7]. Unfortunately, secondary homocysteine lowering in a large randomized clinical trial did not improve survival or reduce the incidence of vascular disease in patients with advanced chronic kidney disease or end-stage kidney disease [43].

Acylcarnitines, metabolites of carnitine, are inversely correlated with blood pressure and cholesterol levels in hemodialysis patients [44]. In another murine model an increase in the levels of acylcarnitines was detected after ischemia-reperfusion injury of the kidney together with associations of certain bacterial species and metabolite levels (*Rothia* and *Staphylococcus* abundance positively correlated with severity of AKI) [10]. The exact mechanism is still unknown but suggests potential targets for future therapeutic interventions. 

Hippurate on the other hand is reported by some to be a metabolomic marker of gut microbiome diversity [45]. In a 1994 study of 35 kidney transplant recipients, hippuric acid levels were significantly higher in patients with acute allograft rejection. It was hypothesized that this is due to reduced excretion; blood levels fell after successful antirejection treatment and microbiota composition was not assessed [12] but it might be another suggestion warranting further research.

### 4.3. Potential Interventions

Simple nutritional interventions, that modify gut microbiota, might lower uremic toxin production. Oral adsorbents, prebiotics, probiotics, synbiotics, eubiotics or fecal microbiota transplantation (FMT) are possible therapeutic options (Figure 3).

One strategy to decrease gut-derived uremic toxins is adsorbent use. Oral uremic toxin adsorbents are present on the market and have been proven to work. AST-120 (marketed as Kremezin^®^) binds many low-molecular-weight compounds with superior adsorption of p-cresol sulfate and indoles. In some human phase II and phase III studies it slowed CKD progression, but unequivocal results have not been demonstrated [46,47]. In a murine model, treatment with AST-120 reduced indole levels and decreased renal expression of mRNAs of injury-related markers [19].

Prebiotics are defined as “selectively fermented ingredients that result in specific changes in the composition and/or activity of the gastrointestinal microbiota, thus conferring benefit(s) upon host health” [48]. The role of different prebiotics in CKD has been studied with promising results; in a prospective crossover single-blind study gum arabic fiber combined with low-protein diet patients had lower serum urea nitrogen levels [49]. We have not found any studies on the use of prebiotics in AKI; a trial NCT03877081 on the effect of probiotics and prebiotics on renal function in septic acute kidney injury patients is registered on ClinicalTrials.gov [50]. On the other hand, anaerobic fermentation of prebiotics results in the production of short chain fatty acids: acetate, butyrate and propionate. SCFAs are recognized as potential mediators modifying intestinal immune function (they modulate inflammatory gene expression, chemotaxis, cell differentiation, proliferation and apoptosis and also improve the course of many inflammatory diseases) [51]. Many experimental studies in mice have shown that SCFAs (especially acetate and butyrate) administration improves outcomes of acute kidney injury [16,17,18,22]. The potential mechanisms need more elucidation, but as a decrease in apoptosis and increase in autophagy together with an increase in mitochondrial DNA content have been shown [16,18], it might be energy conservation and improvement of mitochondrial energetics. The role of olfactory receptor 78 (olf78), a type G protein-coupled receptor that is expressed in the juxtaglomerular apparatus and modulates renin secretion in response to SCFAs, is underlined in some studies [52].

Probiotics are “live microorganisms that, when administered in adequate amounts, confer a health benefit on the host” [53]. Small experimental studies demonstrate that *Lactobacillus salivarius* or a microbial cocktail may ameliorate acute kidney injury [23,24]. This seems to be in contrast to studies showing that broad-spectrum antibiotics protected against AKI [20], although further elucidation is needed, as we do not know the exact microbiota content after antibiotic treatment (a possibility exists that protective bacteria genres were spared).

Synbiotics are nutritional supplements that combine both prebiotics and probiotics. In small human CKD studies showed lower concentrations of p-cresol but not indoxyl sulfate after synbiotic use, although results are inconclusive [54,55]. Studies in AKI are lacking.

Substances, which induce homeostatic changes in the intestinal microbiota or eubiosis are sometimes called eubiotics. Rifaximin–a poorly absorbed antibiotic lowered TMAO levels in mice [56] but in a randomized control trial in CKD patients failed to lower TMAO, p-cresol sulfate, indoxyl sulfate, kynurenic acid, deoxycholic acid, and inflammatory cytokines but did decrease bacterial richness and diversity [57]. A retrospective study that analyzed patients with hepatic cirrhosis found a reduced incidence rate of AKI and hepatorenal syndrome and a lower need for renal replacement therapy [25].

Fecal microbiota transplant is widely accepted as definitive treatment of *Clostridioides difficile* infection [58]. A recently completed trial NCT04361097 registered on ClinicalTrials.gov has evaluated fecal microbiota transplantation in CKD, results are as yet unknown [59], another similar trial is currently recruiting [60]. As yet, there is no data on FMT in AKI, but it remains a possible therapeutic strategy.

## 5. Conclusions

In the beginning of the 20th century, Ilya Ilyich Metchnikoff hypothesized that the human colon functioned only as a waste reservoir, where microbiota produced only “fermentations and putrefaction harmful to the host” [61]. Today the paradigm is completely changing. Increased data provide evidence of existence of a gut-kidney axis in acute kidney injury, mediated in part by uremic toxins. Experimental studies provide concepts that modification of the microbiota composition may result in better AKI outcomes. Small interventional studies in animal models and in humans show promising results, therefore microbiome-targeted therapy for AKI treatment is a promising possibility.

## 6. Materials and Methods

We undertook a systematic search of MEDLINE through PubMed and COCHRANE LIBRARY database from inception to October 10, 2020. 

We searched both with individual keywords and Medical Subject Headings (MeSH) with all subheadings included. Individual keywords used were “acute kidney injury”, “acute renal failure”, “microbiota”, “microbiome”, “uremic toxins”, “gut-derived toxins”, “nephrotoxicity”. MeSH terms included were “acute kidney injury”, “microbiota”, and “uremia middle molecule toxins”. The terms were combined individually using the Boolean operator “AND”. In addition, the references of eligible papers were searched manually for additional records. The results were merged together with duplicates discarded. Remaining articles were screened for relevance (based on their title, abstract, or full text). All articles on the subject of microbiota and gut-derived uremic toxins in the course of acute kidney injury were included due to paucity of data: original studies and reviews/perspectives were both experimental and human. Articles were excluded only if they were clearly related to other subject matters or were not published in English, French, or German. Additionally, ClinicalTrials.gov was searched with simple keywords (kidney, microbiome, microbiota, gut-derived) for currently registered and active studies.

## Figures and Tables

**Figure 1 toxins-12-00788-f001:**
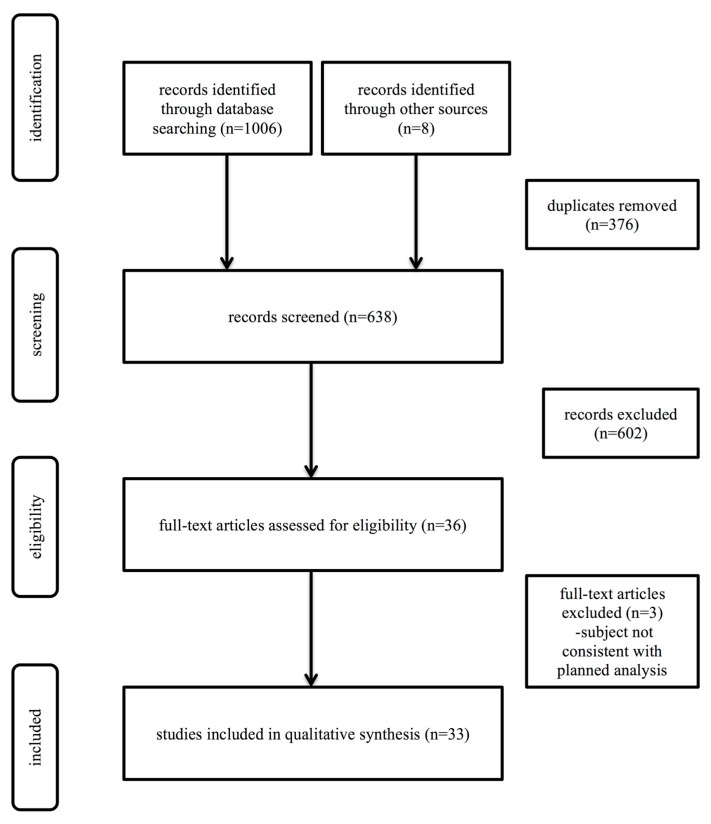
Search strategy and results.

**Figure 2 toxins-12-00788-f002:**
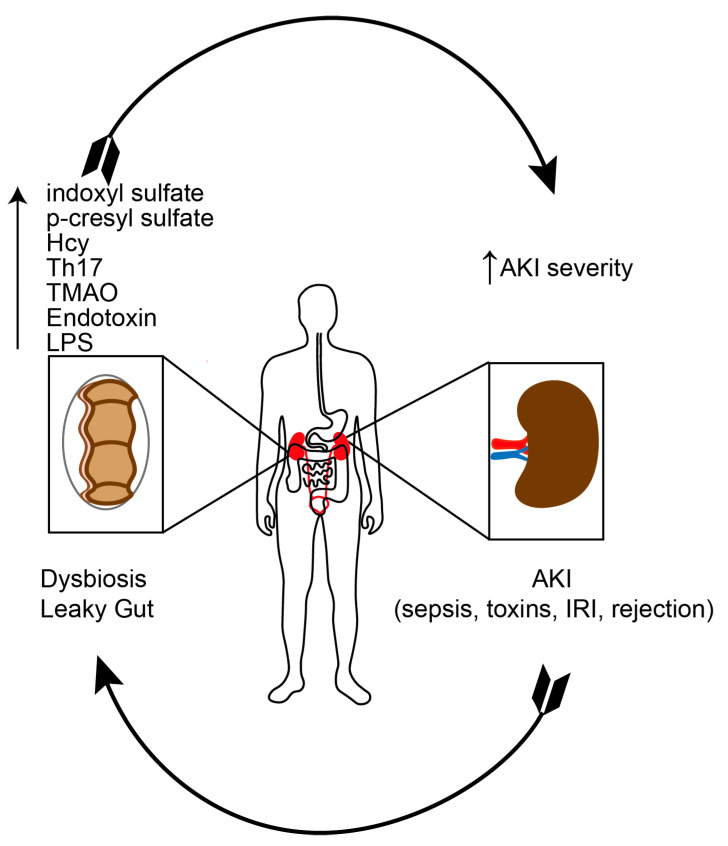
Gut-kidney axis in acute kidney injury (AKI, acute kidney injury; Hcy, homocysteine; IRi, ischemia-reperfusion injury; LPS, lipopolysaccharide; TMAO, trimethylamine N-oxide).

**Figure 3 toxins-12-00788-f003:**
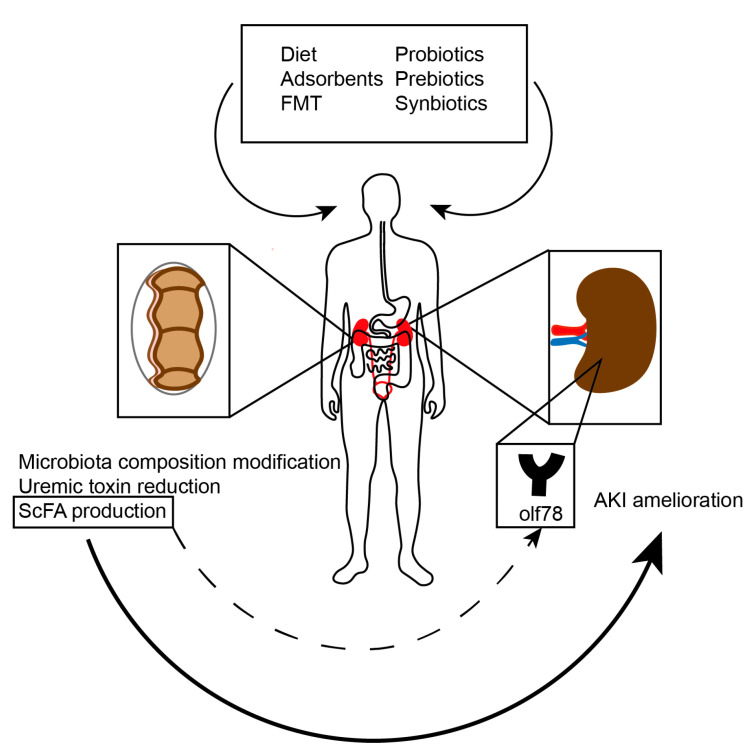
Potential microbiota-targeting interventions in acute kidney injury (AKI, acute kidney injury; FMT, fecal microbiota transplantation; OLF 78, olfactory receptor 78; ScFAs, short chain fatty acids).

**Table 1 toxins-12-00788-t001:** Original studies assessing the relationship between microbiome-derived toxins and acute kidney injury.

Author	Year	Population	Uremic Toxin/Parameter Assessed	Outcome	Results and Key Observations
			Experimental		
Jang et al. [5]	2009	germ-free mice, afterwards fed bacteria-rich diet	numbers and phenotypes of T cells and NK cells, panel of cytokines	extent of renal injury and functional decline after IRI	microbial stimuli influence the phenotype of renal lymphocytes and ameliorate the extent of renal injury
Samanta et al. [6]	2017	Wistar rats	microbiota composition	hypoxia induced AKI	hypobaric hypoxia causes both AKI and affects gut microbial population
Long et al. [7]	2017	C57BL/6 mice	influence of elevated Hcy levels	cisplatin-induced AKI	cisplatin induces more severe tubular injury, tubular cell apoptosis and lower proliferation in hyperHcy mice
Li et al. [8]	2019	Sprague-Dawley rats	gut-derived endotoxin	increased renal mRNA of TLR4 and proinflammatory mediators (Il-6 and MCP-1)	endotoxin increases intrarenal inflammatory response
Yang et al. [9]	2020	C57BL/6 mice and germ-free C57BL/6 mice	microbiota composition	severity of IRI	intestinal dysbiosis, inflammation and leaky gut are consequences of AKI but also determine its severity
Andrianova et al. [10]	2020	Wistar rats	microbiota composition, levels of selected toxins (acylcarnitines)	severity of IRI, creatinine and urea levels	specific bacteria in the gut may ameliorate or aggravate IRI and affect toxin levels
Mishima et al. [11]	2020	germ-free mice and mice with microbiota	metabolome analysis	extent of kidney damage in adenine-induced AKI	germ-free mice enhanced host purine metabolism and exacerbated kidney damage
			Human		
Knoflach et al. [12]	1994	retrospective	hippuric acid concentration	acute kidney allograft rejection	hippuric acid concentration was higher in patients with acute allograft rejection and fell after antirejection treatment
Carron et al. [13]	2019	prospective observational design: 146 kidney transplant recipients	circulating lipopolysaccharide	chronic inflammation and acute rejection episodes	chronic exposure to LPS in the period before transplantation can promote endotoxin tolerance and those patients are less prompt to develop acute rejection after transplantation
Wang et al. [14]	2019	prospective observational design: 262 patients with hospital-acquired AKI	serum indoxyl sulfate levels	90-day mortality	serum indoxyl sulfate levels were elevated in patients with AKI and associated with a worse prognosis
Veldeman et al. [15]	2019	prospective observational design:194 patients with sepsis	serum indoxyl sulfate and p-cresyl sulfate levels	acute kidney injury due to sepsis	serum indoxyl sulfate and p-cresyl sulfate levels were higher in patients with AKI and correlated with AKI course

Abbreviations: AKI, acute kidney injury; Hcy, homocysteine; Il-6, interleukin-6; IRI, ischemia-reperfusion injury; LPS, lipopolysaccharide MCP-1, monocyte chemoattractant protein-1; mRNA, messenger ribonucleic acid; TL4, toll-like receptor 4.

**Table 2 toxins-12-00788-t002:** Original studies assessing microbiome-modifying interventions on acute kidney injury.

Author	Year	Population	Uremic Toxin/Parameter Assessed	Outcome	Results and Key Observations
			Experimental		
Machado et al. [16]	2012	Wistar rats	SCFA (sodium butyrate)	levels of creatinine, inflammatory markers and histology in contrast induced AKI	SCFA treatment attenuated creatinine levels and histological damage
Sun et al. [17]	2013	Sprague-Dawley rats	SCFA (sodium butyrate)	levels of creatinine, AKI markers, antioxidant enzymes and histology in gentamicin induced AKI	chronic treatment with SCFA protects from gentamicin-induced nephrotoxicity
Andrade-Oliveira et al. [18]	2015	C57BL/6 mice	SCFAs (acetate, butyrate, propionate)	levels of creatinine and urea, necrosis score in kidney tubular epithelial cells in IRI	mice treated with acetate-producing bacteria had improved mitochondrial biogenesis and better outcomes
Fujii et al. [19]	2016	SH rats	AST-120	myocardial infarction induced kidney damage	treatment with AST-120 may have protective effects (reduced indole levels and urine, serum biomarkers of kidney injury)
Emal et al. [20]	2017	C57BL/6 wild-type mice	broad-spectrum antibiotics	renal damage and tubular integrity after IRI	depletion of gut microbiota protects against renal injury
Nakade et al. [21]	2018	germ-free C57BL/6 mice	D-serine	hypoxia-induced tubular damage and kidney function	renoprotective effects of gut-derived D-serine in AKI proven
Al-Harbi et al. [22]	2018	BALB/c mice	SCFA (sodium acetate)	kidney function/ renal peroxidase activity/kidney tubular structure in sepsis induced AKI	acetate ameliorates sepsis-induced kidney injury by restoration of oxidant–antioxidant balance in T cells
Lee et al. [23]	2020	Sprague-Dawley rats and Caco-2 cells	*Lactobacillus salivarius* BP121	cisplatin-induced AKI occurence	*L. salivarius* BP121 reduced Caco-2 damage and protected against cisplatin-induced AKI
Zheng et al. [24]	2020	BALB/c mice and Bama miniature pigs	microbial cocktail (*Escherichia, Bacillus, Enterobacter*)	urea and creatinine concentration in nephrotoxin-induced AKI (adenine, cisplatin, glycerol)	in both murine and porcine models of AKI the orally delivered cocktail reduced urea and creatinine concentration
			Human		
Dong et al. [25]	2016	retrospective analysis, 176 cirrhotic adult patients (88 treated with rifaximin	rifaximin	AKI & HRS risk	incidence rate ratio of AKI and HRS, as well as the risk of RRT was lower in the rifaximin group

Abbreviations: Abbreviations: AKI, acute kidney injury; HRS, hepatorenal syndrome; IRI, ischemia-reperfusion injury; RRT, renal replacement therapy; SCFAs, short chain fatty acids; SH, spontaneously hypertensive.

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
