# Peer review of "The Links between Microbiome and Uremic Toxins in Acute Kidney Injury: Beyond Gut Feeling—A Systematic Review"

_toxins, 2020, doi:10.3390/toxins12120788_

Round 1

Reviewer 1 Report

  1. Please clarify and detail the inclusion and non –inclusion criteria in the methods section
  2. Several references related to AKI in renal transplant recipients (e.g. Knoflach Transplant Int 1994), and others related to protein-bound uremic toxins and AKI (e.g. Veldeman Int Urol Nephrol 2019) are missing and need to be added
  3. The authors should provide a Figure summarizing the essential messages
  4. The potential toxicities of homocysteine are not confirmed in RCT when the treatment by folate did decrease the levels of homocysteine but did not modify the outcomes. This data should be reported

Author Response

Dear editors and reviewers,

Thank you for your letter and for the reviewers’ comments on our manuscript entitled “The links between microbiome and uremic toxins in acute kidney injury: beyond gut feeling – a systematic review.” (ID:toxins-1002231). All of these comments were very helpful for revising and improving our paper. We have studied these comments carefully and have made corresponding corrections that we hope will meet with your approval. The changes in the revised manuscript are highlighted in yellow. Responses to the reviewers’ comments are provided below.

We would like to express our great appreciation to you and the reviewers for the comments on our paper. If you have any further queries, please do not hesitate to contact us.

Kind regards,

Alicja Rydzewska-Rosołowska

Reviewer 1.

  1. Please clarify and detail the inclusion and non –inclusion criteria in the methods section
  2. Several references related to AKI in renal transplant recipients (e.g. Knoflach Transplant Int 1994), and others related to protein-bound uremic toxins and AKI (e.g. Veldeman Int Urol Nephrol 2019) are missing and need to be added
  3. The authors should provide a Figure summarizing the essential messages
  4. The potential toxicities of homocysteine are not confirmed in RCT when the treatment by folate did decrease the levels of homocysteine but did not modify the outcomes. This data should be reported

Response to reviewer 1.

Thank You for Your comments. We would like to address them all.

  1. We’re sorry if we haven’t made it completely clear what the inclusion and exclusion criteria were for paper selection. All articles on the subject of microbiota and gut-derived uremic toxins in the course of acute kidney injury were included due to paucity of data: original studies and reviews/perspectives both experimental and human. Articles were excluded only if they were clearly related to other subject matter or were not published in English, French or German. Additonal information on data selection was added in the materials and methods section (page 10-11, lines 228-230).
  2. The requested references were added to the discussion (page 8, lines 124-125 & 147-151; Table 1). They were not included initially due to lack of information on microbiota composition but we agree they are completely relevant.
  3. Two figures, one summarizing gut-kidney axis in acute kidney injury and showing potential microbiota-targeting interventions in acute kidney injury were added (Figure 2&3, page 7&10).
  4. Data on lack of homocysteine lowering efficacy was added to the discussion (page 8, lines 138-140).

Reviewer 2 Report

Dear authors, your review about uremic toxins in acute kidney injury is a very interesting and emerging topic. You gathered a lot of information but the overall review presentation could be improved.

General comment:

You should make a couple of figures summarizing 1) the overall gut-kidney axis structure and cross-talk, and 2) The potential interventions including promising targets.

I'm wondering how "systematic" this review is. A quick search in PubMed for the terms "Gut kidney axis uremia" lead me to a review about oral iron supplementation in CKD patients mentioning a colo-renal axis impacting both the gut microbiome and kidney functions (Kortman et al., 2017). In the present review I don't see any reference to oral iron supplementation nor colo-renal axis.

Specific comments:

-Table 1 and table 2: please separate experimental studies from human studies, I mean don't mixed them in the table.

-Line 143: you mentioned FMT but you don't discuss it. Therefore either remove it or include relevant information related to the kidney disease

Author Response

Dear editors and reviewers,

Thank you for your letter and for the reviewers’ comments on our manuscript entitled “The links between microbiome and uremic toxins in acute kidney injury: beyond gut feeling – a systematic review.” (ID:toxins-1002231). All of these comments were very helpful for revising and improving our paper. We have studied these comments carefully and have made corresponding corrections that we hope will meet with your approval. The changes in the revised manuscript are highlighted in yellow. Responses to the reviewers’ comments are provided below.

We would like to express our great appreciation to you and the reviewers for the comments on our paper. If you have any further queries, please do not hesitate to contact us.

Kind regards,

Alicja Rydzewska-Rosołowska

Reviewer 2.

Dear authors, your review about uremic toxins in acute kidney injury is a very interesting and emerging topic. You gathered a lot of information but the overall review presentation could be improved.

General comment:

You should make a couple of figures summarizing 1) the overall gut-kidney axis structure and cross-talk, and 2) The potential interventions including promising targets.

I'm wondering how "systematic" this review is. A quick search in PubMed for the terms "Gut kidney axis uremia" lead me to a review about oral iron supplementation in CKD patients mentioning a colo-renal axis impacting both the gut microbiome and kidney functions (Kortman et al., 2017). In the present review I don't see any reference to oral iron supplementation nor colo-renal axis.

Specific comments:

-Table 1 and table 2: please separate experimental studies from human studies, I mean don't mixed them in the table.

-Line 143: you mentioned FMT but you don't discuss it. Therefore either remove it or include relevant information related to the kidney disease

Response to reviewer 2.

Thank You for Your comments. We would like to address them all.

  1. We completely agree that figures would add immensely to the paper, two figures according to your suggestions were added (Figure 2&3, page 7&10).
  2. We did not include the mentioned reference in the review because we wanted to limit the subject to acute kidney injury and not include any references pertaining to chronic kidney disease that’s why there is no reference to oral iron supplementation or colo-renal axis as it was studied in chronic kidney disease. We think that when it comes to the subject of microbiome and gut-derived uremic toxins in acute kidney injury our review is systematic.
  3. The data in the tables were separated.
  4. FMT was further discussed in line 199-203, page 9.

Round 2

Reviewer 1 Report

The authors did answer correctly to my queries

I have no additional comment

Author Response

Thank You very much for Your comments

Reviewer 2 Report

Dear authors, thank you for your answers.

-Figures are satisfactory, however for figure 3, I would recommend to harmonize the format for the olfactory receptor 78 (OLF 78 vs olf78 in the plain text).

-I'm still not convinced about the systematic approach. In your answer to my previous comments, you wrote that you didn't include any CKD references while you did, ie. ref. 44,45 and 58. In addition, the latter about FMT only refers to CKD not AKI but still you included it in your list of potential target. Why then don't speak about iron supplementation and colon-renal axis?

Author Response

Thank You very much for Your comments.

We have improved the figures and changed OLF78 into olf78 in Figure 3.

We agree that there is some data on CKD in the discussion section - we speculate a little bit, since many informations are lacking. Therefore on page 7, lines 103-107 we have added a passage on the role of oral iron in kidney-gut axis.